# Cell-Based Models of ‘Cytokine Release Syndrome’ Endorse CD40L and Granulocyte–Macrophage Colony-Stimulating Factor Knockout in Chimeric Antigen Receptor T Cells as Mitigation Strategy

**DOI:** 10.3390/cells12212581

**Published:** 2023-11-06

**Authors:** Ala Dibas, Manuel Rhiel, Vidisha Bhavesh Patel, Geoffroy Andrieux, Melanie Boerries, Tatjana I. Cornu, Jamal Alzubi, Toni Cathomen

**Affiliations:** 1Institute for Transfusion Medicine and Gene Therapy, Medical Center—University of Freiburg, 79106 Freiburg, Germany; ladibas@live.com (A.D.); manuel.rhiel@uniklinik-freiburg.de (M.R.); vidisha.bavesh.patel@uniklinik-freiburg.de (V.B.P.); tatjana.cornu@uniklinik-freiburg.de (T.I.C.); 2Center for Chronic Immunodeficiency (CCI), Medical Center—University of Freiburg, 79106 Freiburg, Germany; 3Ph.D. Program, Faculty of Biology, University of Freiburg, 79104 Freiburg, Germany; 4Institute of Medical Bioinformatics and Systems Medicine, Medical Center—University of Freiburg, 79106 Freiburg, Germany; geoffroy.andrieux@uniklinik-freiburg.de (G.A.); melanie.boerries@uniklinik-freiburg.de (M.B.); 5Faculty of Medicine, University of Freiburg, 79106 Freiburg, Germany; 6German Cancer Consortium (DKTK), Partner Site Freiburg, a Partnership between DKFZ and Medical Center—University of Freiburg, 79106 Freiburg, Germany

**Keywords:** CD19-CAR T cells, CRS, CD40L, GM-CSF, CRISPR-Cas9, knockout

## Abstract

While chimeric antigen receptor (CAR) T cell therapy has shown promising outcomes among patients with hematologic malignancies, it has also been associated with undesirable side-effects such as cytokine release syndrome (CRS). CRS is triggered by CAR T-cell-based activation of monocytes, which are stimulated via the CD40L–CD40R axis or via uptake of GM-CSF to secrete proinflammatory cytokines. Mouse models have been used to model CRS, but working with them is labor-intensive and they are not amenable to screening approaches. To overcome this challenge, we established two simple cell-based CRS in vitro models that entail the co-culturing of leukemic B cells with CD19-targeting CAR T cells and primary monocytes from the same donor. Upon antigen encounter, CAR T cells upregulated CD40L and released GM-CSF which in turn stimulated the monocytes to secrete IL-6. To endorse these models, we demonstrated that neutralizing antibodies or genetic disruption of the *CD40L* and/or *CSF2* loci in CAR T cells using CRISPR-Cas technology significantly reduced IL-6 secretion by bystander monocytes without affecting the cytolytic activity of the engineered lymphocytes in vitro. Overall, our cell-based models were able to recapitulate CRS in vitro, allowing us to validate mitigation strategies based on antibodies or genome editing.

## 1. Introduction

T lymphocytes can be genetically engineered to express a chimeric antigen receptor (CAR), a synthetic receptor designed to recognize a tumor-associated antigen [1,2]. Upon manufacturing, CAR T cells are adoptively transferred to patients and were shown to incite durable responses among patients with hematologic malignancies [3,4]. To date, six CAR T cell therapies have been approved by the European Medicines Agency (EMA) and the U.S. Food and Drug Administration (FDA). On the other hand, the administration of CAR T cell therapy can be accompanied by major toxicities, such as cytokine release syndrome (CRS) [5,6]. The incidence and severity of CRS may depend on the CAR design, the target antigen, and the tumor burden among other factors [7]. For example, CRS was reported more frequently among patients receiving CD19-directed CAR T cells than among patients treated with B-cell mature antigen (BCMA)-targeting CAR T cells (42%–100% vs. 5%, respectively) [7,8]. CRS is characterized by the massive release of proinflammatory cytokines, including interleukin (IL)-10, interferon (IFN)-γ, tumor necrosis factor (TNF-α), granulocyte–macrophage colony-stimulating factor (GM-CSF), MCP-1, and particularly IL-6, from immune cells [1,7]. The increase in proinflammatory cytokines levels results in systemic inflammatory responses, i.e., flu-like symptoms, hypotension, hypoxia, and—as a worst-case scenario—multi-organ system failure [8].

The pathophysiological foundation of CRS is the activation of CAR T cells upon antigen recognition, which is a prerequisite for proliferation of CAR T cells and effective killing of the tumor cells [6]. There is a positive correlation between the tumor burden and the severity of CRS. In addition, the induction time of CRS is relatively short, observed within days after CAR T cell infusion into patients [9]. The activated CAR T cells release cytokines that stimulate bystander immune cells, mainly monocytes and macrophages [6]. The activation of these myeloid cells is triggered on the one hand in a cell-to-cell, independent manner: secreted factors include GM-CSF, IFN-γ, and TNF-α [6]. Activation can also occur in a cell-to-cell dependent manner through the CD40 ligand (CD40L), which is upregulated on the surface of the activated CAR T cells, binding to the CD40 receptor (CD40R) expressed on the myeloid cells [6,10]. The activation of monocytes and macrophages in turn results in secretion of IL-1 and IL-6, which are well-known proinflammatory cytokines [10,11]. The massive release of IL-1 and IL-6 leads to a systemic inflammatory response, including endothelial injury and vascular leakage in many organs [6]. Moreover, the activation of endothelial cells can result in a breakdown of the blood–brain barrier (BBB), which allows cytokines and CAR T cells to access the central nervous system (CNS) [6,8].

The management of CRS in mild cases follows supportive care treatment by administering antipyretics. In moderate and severe CRS cases, patients are treated with the IL-6 receptor blocking antibody Tocilizumab. While administration of Tocilizumab leads to rapid resolution of CRS symptoms within hours, Tocilizumab shows poor penetration across the BBB, which does not stop the development of neurotoxicity [6,12]. Along with Tocilizumab, corticosteroids that cross the BBB can be administered. However, the prolonged administration of corticosteroid ablates the CAR T cells’ anti-leukemic effect [13]. As both Tocilizumab and corticosteroids are considered symptomatic treatments, alternative strategies that prevent the onset of CRS instead of treating its symptoms are needed. 

The challenge in modeling CRS is to recapitulate the interplay between engineered CAR T cells and host immune cells in a relatively short period of time. Based on the above-mentioned clinical observations, CD19 appeared to be a suitable target antigen for the evaluation of CAR T-cell-associated CRS. Some CRS mouse models that involved, among others, the reconstitution of immunodeficient mice with human hematopoietic stem and progenitor cells (HSPCs) have been described [10,11,14]. While they are able to reproduce several aspects of CRS-associated toxicity in an organism, they are complex and not amenable to screening strategies. Cell-based in vitro models that recapitulate key physiological aspects of the in vivo situation will be able to bridge the gap between pre-clinical animal models and humans in clinical trials. Moreover, in vitro models may help to identify novel drug targets and reduce the use of animal models, thereby decreasing the duration and cost of drug development [15]. In this study, we aimed at establishing two robust cell-based CRS in vitro models and used it to validate the hypothesis that disrupting the *CD40L* and/or *CSF2 loci* in CAR T cells using CRISPR-Cas9 will reduce monocyte activation without affecting the antitumor activity of the engineered T cells. 

## 2. Materials and Methods

### 2.1. Cultivation of Cells

Raji cells (DSMZ, Braunschweig, Germany, cat# ACC319), NALM6 (DSMZ, cat# ACC128), and K562 cells (DSMZ, cat# ACC10) were cultivated in RPMI-1640 (GIBCO, Life Technologies, Waltham, MA, USA, cat# 11875093), supplemented with 10% fetal calf serum (FBS, PAN-Biotech, Aidenbach, Germany, cat# P40-47500) and 1% penicillin-streptomycin (P/S, Sigma-Aldrich, St. Louis, MO, USA, cat# P0781). HEK293T cells (ATCC, Manassas, VA, USA, cat# CRL-3216™) were cultured in DMEM high glucose GlutaMAX medium (GIBCO, Life Technologies, Waltham, MA, USA, cat# 61965-026) supplemented with 10% FCS, 1% P/S and 1% Sodium Pyruvate (Biochrom, Cambridge, United Kingdom, cat# L0473). Peripheral blood mononuclear cells (PBMCs) were isolated by gradient centrifugation (Ficoll-Paque) from a leukocyte reduction system (LRS) chamber obtained from healthy donors (informed consent, Blood Donation Center of the Medical Center—University of Freiburg, Freiburg, Germany). PBMCs and primary T cells were cultivated in T cell medium (RPMI-1640 supplemented with 10% FCS, 1% P/S, 10 mM HEPES (Sigma-Aldrich, St. Louis, MO, USA, cat# H3375), 100 U/mL IL-2 (ImmunoTool, Friesoythe, Germany, cat# 11340027), 25 U/mL IL-7 (Miltenyi Biotec, Bergisch-Gladbach, Germany, cat# 130-095-361), and 50 U/mL IL-15 (Miltenyi Biotec, Bergisch-Gladbach, Germany, cat# 130-095-762)). All cells were incubated at 37 °C at 5% CO_2_.

### 2.2. Production and Enrichment of CD19-Targeting CAR T cells

PBMCs were thawed and activated with ImmunoCult^TM^ (human CD3/CD28/CD2, STEMCELL, Vancouver, BC, Canada, cat# 10970) one day later. For transduction, 1 × 10^6^ cells were seeded in wells of a 48-well plate in 500 μL T cell medium supplemented with 5 μg/mL protamine sulfate. CD19-CAR coding lentiviral vector [16] was added at a dose of ~100 transducing units (TU)/cell (based on titration on Jurkat cells), and plates were centrifuged at 1600× *g* for 90 min at 37 °C. Two days later, cells were electroporated (4D-Nucleofector, kit P3 (Lonza, Basel, Switzerland, cat# V4XP-3032), program EO-115) to deliver CRISPR-Cas9 ribonucleoproteins (RNPs), which were pre-formed for 10 min at room temperature at a 1:3 molar ratio (18.3 pmol SpCas9 protein (IDT, Coralville, IA, USA, cat# 1081059) and 55 pmol gRNA (Biolegio, Nijmegen, The Netherlands)). When generating double knockout CAR T cells, 4 nucleases were delivered, each RNP at a dose of 9.15 pmol Cas9 and 28 pmol gRNA (Table 1). Afterwards, the cells were cultured in 200 μL fresh media and supplemented with IL-2 (1000 U/mL for the first 24 h, then 100 U/mL), IL-7 (25 U/mL), and IL-15 (50 U/mL) in a 96 well U-shaped bottom (Corning, Corning, NY, USA, cat# 353077). Where indicated, CAR T cells were enriched using a CD271 MicroBead Kit (Miltenyi Biotec, Bergisch-Gladbach, Germany, cat# 130-099-023). Post-CD271 selection, the CAR T cells were recovered for at least 18 days before using them in an in vitro assay. Where indicated, T cells were activated with 20 ng/mL Phorbol-12-myristat 13-acetat (PMA; Sigma-Aldrich, St. Louis, MO, USA, cat# P8139) and/or 1 μg/mL ionomycin (Sigma-Aldrich, St. Louis, MO, USA, cat# 407952).

### 2.3. Genotyping and Assessment of Off-Target Effects

The DECODR analysis [17] was performed using primers listed in Table 2. Chromosomal aberrations arising from on-target and off-target activities of the CRISPR-Cas9 nucleases were detected by CAST-Seq as previously described [18,19], using primers defined in Table 2. CAST-Seq data was analyzed using a revised bioinformatics pipeline optimized for concomitant delivery of two RNPs (Klermund et al., submitted for publication). Detailed results from all CAST-Seq analyses are provided in Appendix A.

### 2.4. Flow Cytometry

The following antibodies and fluorochromes were used: CD19CAR Detection Reagent (Miltenyi Biotec, Bergisch-Gladbach, Germany, cat# 130-115-965), anti-Biotin-PE (clone REA746, Miltenyi Biotec, cat# 130-110-951), anti-Biotin-FITC (Miltenyi Biotec, cat# 130-090-857), anti-human CD3-APC (clone BW264/56, Miltenyi Biotec, cat# 130-113-125), anti-human CD14-FITC (clone RMO52, Beckman-Coulter, Brea, CA, USA, cat# B36297), anti-human CD19-FITC (clone H1B19, BioLegend, San Diego, CA, USA, cat# 302206), anti-human CD25-PE (clone 4E3, Miltenyi Biotec, cat# 130-113-282), anti-human CD25-APC (clone 4E3, Miltenyi Biotec, cat# 130-113-280), anti-human CD69-PE (clone FN50, BD, Heidelberg, Germany, cat# 555531), anti-human CD40-PE (clone 17, Sino Biological, Eschborn, Germany, cat# 10774-MM17-P), anti-CD154 (CD40L)-FITC (clone 24-31, Invitrogen, Waltham, MA, USA, cat# 11-1548-42), anti-CD271 (LNGFR)-APC (clone REA648, Miltenyi Biotec, cat# 130-116-497), anti-human GM-CSF-PE (clone BVD2-21C11, BD, Heidelberg, Germany, cat# 554507), anti-human Fcɣ Block (BD, Heidelberg, Germany, cat# 564220), anti-human mouse-IgG2a-FITC (BD, Heidelberg, Germany, cat# 555573), and anti-human mouse-IgG1,K-PE (BD, Heidelberg, Germany, # 556650). To evaluate live/dead cells, propidium iodide (PI, Sigma-Aldrich, St. Louis, MO, USA, cat# P4170) was used. DPBS (Thermo Fisher Scientific, Waltham, MA, USA, cat# 14190250) was used to wash the cells. The FACS buffer contained DPBS with 5% FBS, 0.1% Na^+^-azide and 2 mM EDTA. All samples were analyzed on a Accuri C6 flow cytometer (BD, Heidelberg, Germany). Data were analyzed using FlowJo V10 (BD, Heidelberg, Germany). 

### 2.5. Cytolytic Activity of CAR T Cells

The 1 × 10^4^ CD19-targeted CAR T cells (effector cells) and 1 × 10^4^ CFSE-labelled Raji cells (CD19^+^ target cells) were co-cultured at a 1:1 or 2.5:1 ratio in 200 μL of RPMI-1640 supplemented with 10% FCS, 1% P/S and 10 mM HEPES in 96 well plates with U-shaped bottom. As a control, non-transduced T cells were used. After 48 h, cytolytic activity was calculated through determining the fraction of live CFSE^+^ target cells through flow cytometry (Accuri C6, BD, Heidelberg, Germany). Supernatants were collected to determine the concentration of released cytokines. 

### 2.6. CRS In Vitro Model 

Monocytes were isolated from PBMCs of the same donor used to produce the CAR T cells using EasySep^TM^ human CD14 positive selection kit II (STEMCELL, Vancouver, Canada, cat# 17858). The 1 × 10^5^ monocytes were seeded either in wells of a 24-well plate (Sarstedt, Nümbrecht, Germany) or in a transwell insert with 0.4 μm pore size (Corning, Corning, NY, USA, cat# 3470) in case cell-to-cell independent activation was evaluated. After 4–6 h, gene-edited CD19-CAR T cells and CD19^+^ Raji cells were added at a 1:1:1 ratio in a total of 1 mL RPMI-1640 supplemented with 10% FCS and 1% P/S. As a control CD19^−^ K562 cells were used as target cells. Where indicated, anti-human CD40L neutralizing antibody (InvivoGen; San Diego, CA, USA cat# mabg-h40l-3) and/or anti-human GM-CSF neutralizing antibody (BD, Heidelberg, Germany, cat# 554502) was added. Supernatants were collected after 72 h to measure cytokine concentration.

### 2.7. Cytokine Concentrations

Cytometric Bead Array (CBA) was used to determine the concentration of IFN-γ, GM-CSF, Granzyme B, TNF-α, IL-10, and IL-6 (BD, Heidelberg, Germany, cat# 560111, 558335, 560304, 560112, 558274, and 558276, respectively) following the manufacturer’s instructions. Data were acquired using a BD FACSCanto II flow cytometer (BD, Heidelberg, Germany) and were analyzed using FlowJo V10 (BD, Heidelberg, Germany).

### 2.8. Statistics

All statistical analyses were performed using GraphPad 8.4.3 (GraphPad, San Diego, CA, USA) using one-way Anova test or Dunnett’s multiple comparisons test, as indicated in each figure.

## 3. Results

### 3.1. Modelling CRS In Vitro

To investigate the potential impact of mitigation strategies aiming at reducing CRS, we established two CRS in vitro models that focused on the interplay between CD19^+^ tumor cells, CD19-targeted CAR T cells, and bystander monocytes. The principle of these in vitro models is based on CD19 antigen bearing tumor cells activating CD19-targeted CAR T cells, which in turn stimulate the bystander monocytes to secrete proinflammatory cytokines. To simplify the assay, we used the Raji cell line as a CD19^+^ target cell type. Conversely, to avoid excessive alloreactivity of the primary T cells, we made sure that both the CD19-targeting CAR T cells as well as the monocytes derived from the same healthy donors. The CD19-targeted CAR T cells were manufactured by lentiviral transduction using a vector coding for a 2nd generation CD19-CAR that consisted of a CD19-directed domain (FMC63) with 4-1BB costimulatory and a CD3ζ signaling domain [16]. The CAR construct was linked to the selection marker ∆LNGFR (a truncated form of the low-affinity nerve growth factor receptor) via a P2A peptide, which allowed us to enrich the transduced T cells using CD271-based selection. 

To identify the optimal ratio between the three cell types, we initially determined CAR T cell activation levels induced by either of two CD19^+^ leukemia cell lines, NALM6 and Raji (Appendix A). To this end, we assessed CD69 and CD40L surface expression levels along with the release of IFN-γ and GM-CSF from the activated CAR T cells. The optimal effector/target (E:T) ratio resulting in high CAR T cell activation was identified by co-culturing the CD19-targeting CAR T cells with the CD19^+^ target cell types in either a 1:1 or a 1:10 E:T ratio. When using NALM6 cells, a 1:10 ratio resulted in a higher activation, while a 1:1 E:T ratio resulted in high CAR T cell activation when using Raji cells (Appendix A). Those two E:T ratios were fixed and primary monocytes derived from the same donor were added at a 1:1 ratio with the CAR T cells (Appendix A). We observed that NALM6 cells stimulated the release of IL-6 from monocytes even when co-cultured with untransduced T cells, most likely due to alloreactivity. A further increase in IL-6 secretion was not observed when CD19^+^ NALM6 cells and monocytes were co-cultured with CD19-targeted CAR T cells (Appendix A). On the other hand, the co-culture of CD19^+^ Raji cells with CAR T cells and monocytes at a 1:1:1 ratio induced IL-6 secretion at substantially higher levels when compared to the co-culture experiment with untransduced T cells (Appendix A). Based on these results, we used Raji cells in all subsequent experiments. To validate this CRS in vitro model, we added different amounts of neutralizing CD40L or GM-CSF antibodies to the assay. Compared to the control without antibodies, blocking CD40L with the neutralizing antibody resulted in ~20% reduction in IL-6 levels in the supernatant, while GM-CSF neutralization led to a significant decrease in IL-6 secretion of ~30% (Figure 1A). Using both neutralizing antibodies simultaneously did not further reduce IL-6 levels. IL-6 secretion was dependent on the presence of the CAR, since in the co-culture experiment with untransduced T cells only minimal levels of IL-6 were detected (Figure 1A). 

To better characterize the roles of CD40L and GM-CSF, we separated the monocytes from the CAR T cells by a membrane with a pore size of 0.4 µm, which enables soluble factors, such as GM-CSF, to pass the membrane but not cells (Figure 1B). In this scenario, addition of neutralizing GM-CSF antibodies resulted in ~80% reduction in IL-6 levels. As expected, this effect was not further increased upon adding CD40L neutralizing antibodies. In the transwell setup too, IL-6 secretion was strictly dependent on the presence of a CAR (Figure 1B). 

To evaluate whether neutralizing antibodies have an effect on the cytolytic activity of the CAR T cells, the engineered lymphocytes were co-cultured with carboxyfluorescein succinimidyl ester (CFSE)-labelled CD19^+^ Raji at an E:T ratio of 1:1 and 2.5:1 in the presence or absence of neutralizing CD40L and/or GM-CSF antibodies. Cytolytic activity of the CAR T cells was computed by quantifying the depletion of CFSE^+^ Raji cells over time. As evident from Figure 1C, the antibodies did not have a significant impact on the antitumor activity of the CD-19 targeted CAR T cells. In conclusion, CD40L and/or GM-CSF neutralization using blocking antibodies was able to reduce IL-6 release from bystander monocytes in vitro without impairing the antitumor activity of the CAR T cells.

### 3.2. Disruption of CD40L and/or CSF2 in CAR T Cells

To validate the two targets, we sought to disrupt the genes coding for CD40L and GM-CSF in CAR T cells using CRISPR-Cas9 technology. To achieve efficient disruption, a so-called “double hit” strategy was applied by utilizing two targeted nucleases per locus simultaneously (Figure 2A,B). The Cas9 protein was complexed with the individual gRNA targeting either *CD40L* exon 1 or *CSF2* exon 3, respectively, and transferred to CAR T cells using nucleofection. After eight days, the edited CAR T cells were subjected to phenotype and genotype analyses. The expression of CD40L and GM-CSF was measured by flow cytometry 4 or 24 h after activation with PMA/ionomycin, respectively. The evaluation of the edited CAR T cells confirmed efficient knockout with >80% reduction in CD40L-positive CAR T cells upon transfer of the CD40L-targeting CRISPR-Cas9 nucleases, and >70% reduction in GM-CSF^+^ cells upon editing with the CSF2-targeting designer nucleases (Figure 2C,D). The efficient knockout was maintained when disrupting both *CD40L* and *CSF2* simultaneously to produce the Double^KO^ CAR T cells The flow cytometric evaluation showed >70% reduction in CD40L^+^ cells and almost 60% reduction in GM-CSF^+^ cells (Figure 2C,D). Genotyping of the edited CAR T cells by sequence analysis of the target site using DECODR [17] confirmed the formation of insertion/deletion mutations (InDels) at the target sites (Figure 2E,F). Of note, the phenotypic knockout frequencies were well in line with the genetic analysis. The slightly lower knockout efficiency in the Double^KO^ CAR T cells is related to the fact that in the Double^KO^ samples, smaller amounts of RNP per target locus could be applied as compared to the single KOs (see Materials and Methods). Taken together, the generation of CD40L^KO^ and/or GM-CSF^KO^ CAR T cells using the CRISPR-Cas9 platform was successful with efficient disruption of the two loci, both individually and in combination. 

### 3.3. Assessment of Chromosomal Integrity

Genome editing, in particular multiplexed editing of more than one locus, has been associated with both off-target activity as well as the generation of structural variants, such as chromosomal translocations, inversions and large deletions [20,21,22]. CAST-Seq enables the nomination of off-target sites and allows for a semi-quantitative assessment and classification of chromosomal rearrangements [18,19]. As evident from the CAST-Seq coverage plots, *CD40L*- and *CSF2*-edited cells revealed wide-ranging on-target aberrations, in particular large deletions and inversions spanning a ± 10 kb region around the target sites (Figure 3A,B). Off-target-mediated translocations (OMTs) were not detected in *CD40L*- and *CSF2*-edited CAR T cells, supporting the absence of off-target activity of any of the four nucleases used (Figure 3C,D and Appendix A). To disrupt *CD40L* and *CSF2* simultaneously, the four CRISPR-Cas9 nucleases were delivered concomitantly. CAST-Seq was performed using either of the two target sites as anchors. In line with the previous findings for the single locus edited samples, no OMTs were detected (Figure 3E,F and Appendix A). However, and as expected, simultaneous cleavage of the target sites in *CD40L* and *CSF2* resulted in a high number of translocations between chromosomes 5 and X. In summary, the chosen CRISPR-Cas9 nucleases presented with high knockout efficacy combined with high specificity and the expected chromosomal translocations between the two target sites.

### 3.4. CD40L^KO^ and GM-CSF^KO^ CAR T Cells Eliminate CD19^+^ Tumor Cells

To evaluate the antitumor activity of the gene-edited CAR T cells, the cells were co-cultured with CFSE-labelled CD19^+^ Raji cells at effector/target ratios of 1:1 and 2.5:1, respectively. Cytolytic activity of the gene-edited CAR T cells was calculated through quantifying the depletion of CFSE^+^ Raji cells over time. After 48 h, all CAR T cell samples (i.e., non-edited, CD40L^KO^, GM-CSF^KO^, and Double^KO^ CAR T cells) eliminated some 60% of Raji cells at the 1:1 ratio and some 80% at the 2.5:1 effector/target cell ratio (Figure 4A). Non-transduced T cells only showed cytolytic background activity, likely due to residual alloreactivity. Analysis of the factors released to the supernatant confirmed high activity of the *CD40L*/*CSF2*-edited CAR T cells, with secretion of IFN-γ, granzyme B and GM-CSF upon CD19 target recognition (Figure 4B–D). As a result of efficient *CSF2* disruption, secretion of GM-CSF from GM-CSF^KO^ and Double^KO^ CAR T cells was significantly reduced when compared to non-edited CAR T cells (Figure 4D). In conclusion, *CD40L* and *CSF2*-edited CAR T cells maintained their antitumor activity against leukemic B cells.

### 3.5. CD40L^KO^ and/or GM-CSF^KO^ Mitigates IL-6 Release

To evaluate the impact of CD40L and GM-CSF knockout in the engineered T lymphocytes, CD40L^KO^, GM-CSF^KO^, and Double^KO^ CAR T cells were co-cultured with CD19^+^ Raji cells and monocytes at a 1:1:1 ratio. The release of IL-6 was compared to the co-culture with non-edited CD19-targeted CAR T cells (Figure 5A). Knockout of CD40L or GM-CSF in CAR T cells reduced IL-6 levels in the supernatants by 30–40%. Co-culture with the Double^KO^ CAR T cells decreased IL-6 secretion further to about 50% when compared to non-edited CAR T cells. We did not observe any differences in the release of IFN-γ, TNF-α and IL-10 when comparing the edited CAR T cells with the non-edited CAR T cells (Appendix A). To discriminate between the impact of soluble versus cell-based factors, the monocytes were separated from the CAR T cells in a transwell setting (Figure 5B). GM-CSF knockout resulted in a 60–70% reduction in IL-6 concentration in the supernatant, which was increased to ~90% reduction in the co-culture experiment with the Double^KO^ CAR T cells. As expected, in this setup *CD40L* disruption in CAR T cells did not affect IL-6 release from monocytes. Of note, IL-6 release was dependent on the presence of the CAR in both setups, as in co-culture experiments with untransduced T cells only minimal levels of the cytokine were detected in the supernatant (Figure 5A,B). Furthermore, the co-cultures with CD19-negative K562 tumor cells did not lead to activation of CAR T cells and hence did not stimulate monocytes to release IL-6. In conclusion, the in vitro CRS models allowed us to examine the impact of genome editing-based mitigation strategies, indicating that disruption of both the CD40L and/or GM-CSF encoding loci in CAR T cells reduced IL-6 secretion from bystander monocytes.

## 4. Discussion

CAR T-cell therapies have shown excellent results in treating cancer patients with hematologic malignancies [3,23,24]. However, the therapy’s achievement should not obscure the fact that immune cell therapies may have severe side effects, including on-target/off-tumor toxicity, neurotoxicity, and CRS [6]. CRS occurs when tumor-activated CAR T cells stimulate bystander immune cells, primarily myeloid cells, to release massive amounts of proinflammatory cytokines, particularly IL-6. To prevent and/or treat CRS locally rather than systemically, straightforward cell-based models that simulate the complex in vivo situation can help in identifying key regulators in CRS through medium-throughput screens. We developed two simple and reliable in vitro models to help researchers assess the impact of cellular and/or soluble factors on CRS, specifically the release of IL-6 by bystander monocytes. To validate these CRS models, neutralizing antibodies and gene editing were implemented and shown to reduce IL-6 secretion by bystander monocytes.

GM-CSF secreted from tumor-activated CAR T cells induces proliferation and activation of monocytes and macrophages, leading to an increase in their proinflammatory features [25,26]. Our assays confirmed that GM-CSF secreted from activated CAR T cells is a key soluble factor driving IL-6 secretion from bystander monocytes, and that GM-CSF knockout in CAR T cells can mitigate IL-6 release from bystander monocytes in vitro. This is in line with previous studies reporting that the disruption of the GM-CSF encoding *CSF2* locus in CD19-targeted CAR T cells abolished CRS markers in vitro. Furthermore, the edited CAR T cells showed comparable antitumor activity to the non-edited CAR T cells in vivo and prolonged survival of mice in a NALM6 xenograft model [14,27]. 

It is known that the CD40L-CD40R interaction is important for the activation of macrophages by T cells [6]. This axis has been investigated in several preclinical mouse models. For example, Kuhn et al. used immunocompetent mice to study the additive effect of equipping CD19-directed murine CAR T cells with murine CD40L. They found an enhanced immune response by CD40L-expressing CAR T cells, which allowed the T cells to increase their overall antitumor activity [28]. However, CRS-associated toxicity was not evaluated in these treated mice. Giavridis et al. demonstrated the impact of the CD40L-CD40R interaction in a xenografted immunodeficient mouse model, where the intensity of CRS symptoms was enhanced when CD19-directed human CAR T cells were engineered to express murine CD40L. The interaction of mCD40L with mCD40R expressed on the myeloid cells of these animals significantly increased the levels of murine inflammatory cytokines in the blood, including IL-6 [10]. These data support the notion that CRS is difficult to address in preclinical animal models without genetically engineering human CAR T cells to express murine CD40L. Taken together, these observations underscore the need to develop human cell-based in vitro assays such as the one we have developed. Furthermore, our efforts are in line with the 3Rs (replace, reduce, and refine) principles to replace and/or reduce animal testing. In summary, our data indicate that neutralizing CD40L antibodies reduced the activation of bystander monocytes, making CD40L and CD40R attractive drug targets. We validated this target by demonstrating that a *CD40L* gene knockout in CAR T cells reduced the activation of and IL-6 secretion by co-cultured monocytes without affecting the cytolytic activity of the engineered lymphocytes in vitro. It will be interesting to validate the antitumor activity of *CD40L*-edited CAR T cells in vivo. Thus, our results highlight the role of the CD40 axis in CRS and provide druggable targets for further preclinical and clinical investigation.

In this study, we employed a so-called ‘double hit’ strategy [29], which helped us in achieving high allelic disruption frequencies of ~80% in CAR T cells. Since every DNA double strand break inducing agent poses a threat to genome integrity, a higher knockout efficiency has to be counterbalanced against safety. To evaluate genome integrity and to nominate potential off-target sites of the CRISPR-Cas9 nucleases used in this study, we employed CAST-Seq [18]. CAST-Seq enables genome-wide detection of chromosomal rearrangements induced by on- and off-target activity of designer nucleases, as well as nomination of off-target sites. No off-target activity was detected in any of the CAR T cell samples in which a single locus was targeted by two nucleases, indicating high specificity of the CRISPR-Cas nucleases that we designed. On the other hand, it is well known that targeting two loci simultaneously can induce translocations between the two target chromosomes in more than 1% of CAR T cells [30,31]. Our CAST-Seq data confirmed chromosomal translocations between *CD40L* and *CSF2*. Furthermore, we detected large chromosomal rearrangements at both on-target sites. This phenomenon was previously described by us and by others using the CAST-Seq technology [32,33,34], suggesting that large deletions and inversions at the on-target site occur more frequently than inter-chromosomal translocations.

## 5. Conclusions

We established two simple CRS in vitro models that allowed us to evaluate mitigation strategies. Our data confirmed that disrupting of the GM-CSF encoding *CSF2* locus in CAR T cells decreases the concentration of cytokines released by bystander immune cells. Furthermore, using our assay, we were able to describe for the first time that disruption of the *CD40L* locus in CAR T cells alleviates CRS in vitro. In general, our cell-based CRS assay can be used for the in vitro validation of other druggable targets, for the evaluation of genome editing strategies targeting other players involved in the interplay between the innate immune system and effector (CAR) T cells, and finally for the evaluation of the CRS profile of alternative immune cell types interacting with the myeloid compartment.

## Figures and Tables

**Figure 1 cells-12-02581-f001:**
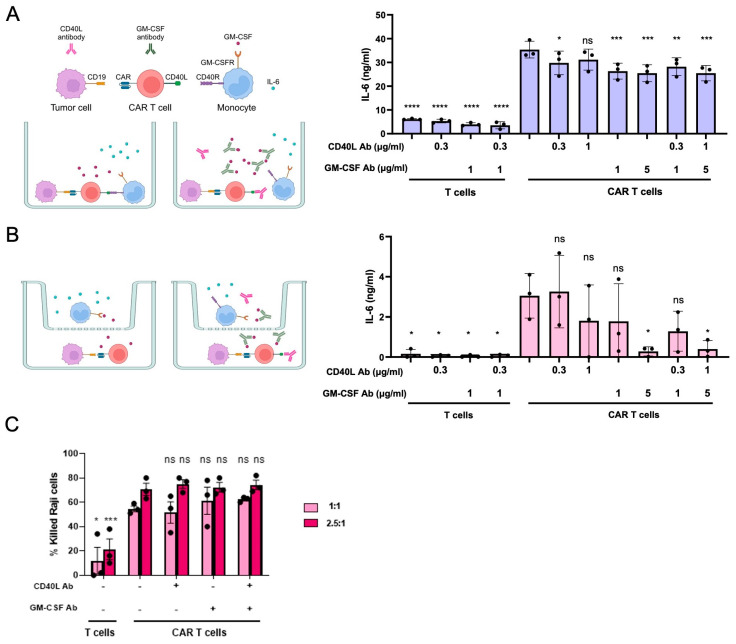
CD40L and GM-CSF neutralization reduces IL-6 release. (**A**) Cell-to-cell-dependent CRS in vitro assay. The schematic of the experimental setup is shown on the left. Depicted is the co-culture of CD19-directed CAR T cells, Raji cells and monocytes in a 1:1:1 ratio in the absence (left) or presence (right) of antibodies that bind to CD40L or GM-CSF. Antibody-mediated neutralization is achieved either by capturing free GM-CSF or by blocking the CD40L-CD40R interaction. IL-6 levels were determined in the supernatant after 72 h. (**B**) Cell-to-cell-independent CRS in vitro assay. The schematic of the experimental setup is shown on the left. Depicted is the co-culture of CD19-targeting CAR T cells and Raji cells in the bottom well and the monocytes in the upper chamber at a 1:1:1 ratio in the absence (left) or presence (right) of CD40L or GM-CSF binding antibodies. IL-6 concentrations in the supernatants were determined after 72 h. (**C**) Cytotoxicity. To detect the fraction of dead cells by flow cytometry after 48 h, CD19^+^ Raji cells were labelled with CFSE prior to co-culture with CAR T cells in the presence or absence of neutralizing antibodies (N = 3). Dunnett’s multiple comparison test with the non-treated CAR T cell group as the control: ns, *, **, *** and **** indicate nonsignificant, *p* ≤ 0.05, *p* < 0.01, *p* < 0.001, and *p* < 0.0001, respectively.

**Figure 2 cells-12-02581-f002:**
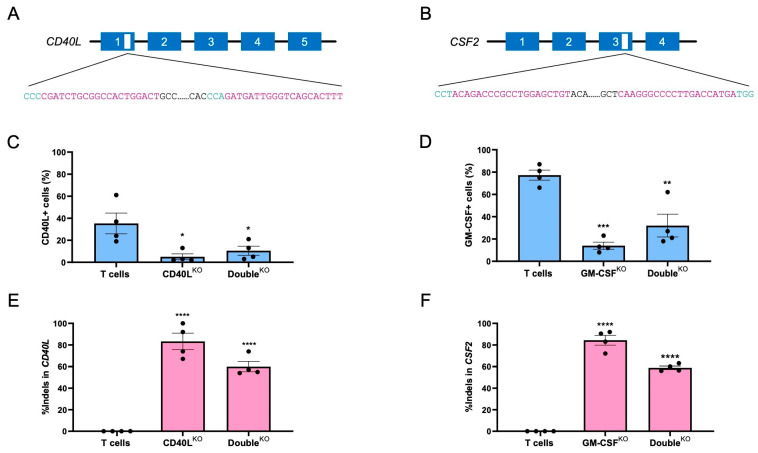
Production of CD40L^KO^ and GM-CSF^KO^ CAR T cells using CRISPR-Cas9 technology. (**A**,**B**) Target sites. Schematically shown are the *CD40L* and *CSF2* loci with the respective numbers of exons and the corresponding CRISPR-Cas9 target sites (protospacer in purple, PAM in green). (**C**,**D**) Phenotype. On day 8 after electroporation expression levels of CD40L (**C**) or GM-CSF (**D**) were determined using flow cytometry 4 or 24 h after activation with PMA/ionomycin, respectively. (**E**,**F**) Genotype. The percentages of Indels at the respective target sites in *CD40L* (**E**) and *CSF2* (**F**) were assessed by DECODR. Data shown are mean and standard error of the mean (SEM) of 4 independent experiments (N = 4). One-way ANOVA test to compare T cells versus edited cells: *, **, ***, and **** indicate *p* < 0.05, *p* < 0.005, *p* < 0.0005, and *p* < 0.0001, respectively.

**Figure 3 cells-12-02581-f003:**
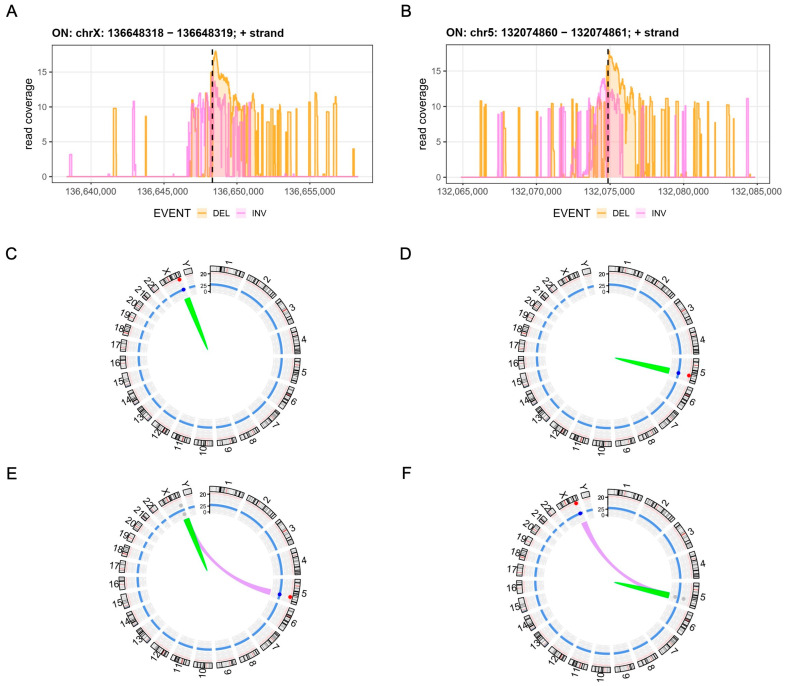
Evaluation of off-target effects. (**A**,**B**) On-target aberrations. Coverage plots show CAST-Seq reads mapped to a ± 10 kb region around the *CD40L* (**A**) or *CSF2* (**B**) target sites. Sequencing direction is from left to right. The x-axis indicates the chromosomal coordinates, the y-axis the log2 read count per million, and the dotted line the cleavage site. Deletions (DEL) are shown in orange; inversions (INV) are shown in purple. (**C**–**F**) Structural variations. Circos plots display the structural variations detected in cells edited at *CD40L* (**C**) or *CSF2* (**D**) individually or simultaneously using either *CD40L* (**E**) or *CSF2 (***F**) as an anchor for CAST-Seq analysis. Aberrations at the on-target site are indicated in green, off-target mediated translocations in red (absent), and translocations between the *CD40L* and *CSF2* target sites in purple.

**Figure 4 cells-12-02581-f004:**
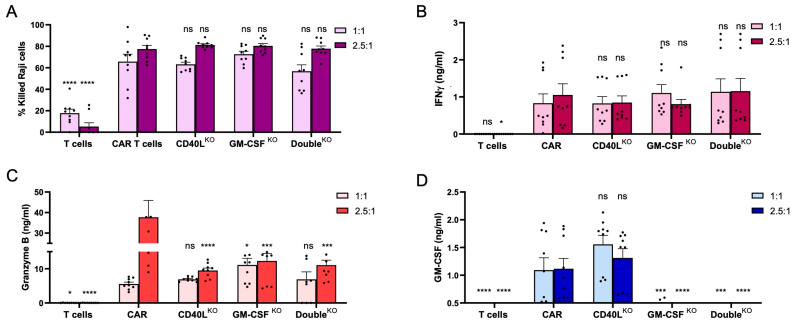
Antitumor activity of *CD40L*- and *CSF2*-edited CAR T cells. CD19-targeting CAR T cells were cultured with CD19^+^ Raji cells at effector/target (E:T) ratios of 1:1 and 2.5:1. (**A**) Cytotoxicity. Raji cells were labelled with CFSE prior to co-culture with CAR T cells to detect the fraction of dead cells by flow cytometry. (**B**–**D**) Cytokine release. Supernatants were collected after 48 h of co-culture and the concentration of IFN-γ, granzyme B and GM-CSF determined using cytometric bead array (N = 3 independent experiments, each in triplicate). One-way ANOVA test with the non-edited CAR T cell group serving as the control: ns, *, *** and **** indicate nonsignificant, *p* < 0.05, *p* < 0.0005 and *p* < 0.0001, respectively.

**Figure 5 cells-12-02581-f005:**
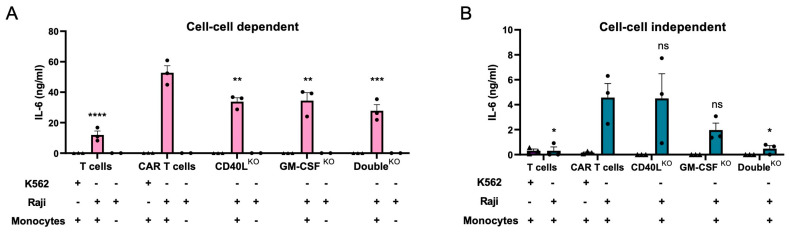
Impact of CD40L and/or GM-CSF knockout in CAR T cells on IL-6 release from bystander monocytes. (**A**) Cell-to-cell dependent effects. Shown are IL-6 levels in the supernatant of the co-culture of CD19-targeting CAR T cells, K562 or Raji cells, and monocytes at a 1:1:1 ratio. (**B**) Cell-to-cell independent. Shown are IL-6 levels in the supernatant of the co-culture of CD19-targeting CAR T cells, K562 or Raji cells, and monocytes in inserts at a 1:1:1 ratio (N = 3). Dunnett’s multiple comparisons test with non-edited CAR T cells serving as the control group: ns, *, **, *** and **** indicate nonsignificant, *p* < 0.05, *p* < 0.005, *p* < 0.0005, and *p* < 0.0001, respectively.

**Table 1 cells-12-02581-t001:** gRNAs binding sequence.

Name	Target Site	Target Sequence (5′-3′)
CD40L-gRNA2	*CD40L* exon 1	AAAGTGCTGACCCAATCATC
CD40L-gRNA3	*CD40L* exon 1	AGTCCAGTGGCCGCAGATCG
CSF2-gRNA2	*CSF2* exon 3	ACAGCTCCAGGCGGGTCTGT
CSF2-gRNA3	*CSF2* exon 3	CAAGGGCCCCTTGACCATGA

**Table 2 cells-12-02581-t002:** Oligonucleotides.

Name	Sequence (5′-3′)
CD40L forward	GGAGAGAAGACTACGAAGCAC
CD40L reverse	GAGACTTCATTGACTAGGCAAC
CSF2 forward	TGACTACAGAGAGGCACAGA
CSF2 reverse	TCACCTCTGACCTCATTAACC
CD40L-decoy reverse	GAAGATACACAGCAAAAAGTGC
CD40L-decoy forward	ATAGAAGGTTGGACAAGGTAAGA
CD40L-bait	GTCTTCTCATGCTGCCTC
CD40L-bait nested	GACTGGAGTTCAGACGTGTGCTCTTCCGATCTGCCACCTTCTCTGCCAGAAGATACC
CSF2-decoy reverse	GCAGTGCTGCTTGTAGTG
CSF2-decoy forward	CTCCAACCCCGGTGAGT
CSF2-bait	TGGTGGAGAGTTCTTGTAC
CSF2-bait nested	GACTGGAGTTCAGACGTGTGCTCTTCCGATCTTGTGGGCACTTGGCCACTG

## Data Availability

Data generated or analyzed during this study are included in this article and its Appendix A. Raw data, including CAST-Seq data and targeted amplicon NGS data, were deposited in the Gene Expression Omnibus repository and are available under GEO accession number GSE243587.

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
