# Peer review of "Cell-Based Models of ‘Cytokine Release Syndrome’ Endorse CD40L and Granulocyte–Macrophage Colony-Stimulating Factor Knockout in Chimeric Antigen Receptor T Cells as Mitigation Strategy"

_cells, 2023, doi:10.3390/cells12212581_

Round 1
Reviewer 1 Report
Comments and Suggestions for Authors
The article " Cells-based models of ‘Cytokine Release Syndrome’ endorse CD40L and GM-CSF knockout in CAR T cells as mitigation strategy " addresses a very interesting topic such as the development of an adverse reaction, the CRS, that can become life-threatening for the patient.
Therefore, the article is interesting, but in my opinion it needs to address some issues before being published.
1 It lacks a cytometric analysis of CD19 expression on Raji and NALM6 cells. Also, why do NALM6 cells not stimulate IL- 6 release from monocytes?
2 An analysis on CAR-T cells targeting other antigens is lacking to confirm the results.
3 The in vitro potency of edited CAR-T cells is comparable to that of WT CAR-T. But I think data from in vivo experiments are also needed to validate these results.
4 What do the authors think about CAR-T cells engineered to express CD40L and enhance immune response? (see Kuhn et al. Cancer Cell. 2019 Mar 18; 35(3): 473-488.e6). The CD40-CD40L axis seems to be implicated in enhancing the cytotoxic response.
5 Finally, to my knowledge CRS is associated, according to a relevant part of the literature, with the presence of a large tumor and thus with the significant post-treatment presence of cells that have gone to death. What do the authors think about this?
Author Response
Comments and Suggestions for Authors
The article " Cells-based models of ‘Cytokine Release Syndrome’ endorse CD40L and GM-CSF knockout in CAR T cells as mitigation strategy" addresses a very interesting topic such as the development of an adverse reaction, the CRS, that can become life-threatening for the patient.
Therefore, the article is interesting, but in my opinion it needs to address some issues before being published.
Response:
We thank the reviewer for his/her positive feedback. Below is our response to the reviewer, including the references to the new datasets provided.
1 It lacks a cytometric analysis of CD19 expression on Raji and NALM6 cells. Also, why do NALM6 cells not stimulate IL- 6 release from monocytes?
Response:
We thank the reviewer for raising these points.
a/ CD19 expression in Raji and NALM6 cells was previously validated by flow cytometry and the data are now included in Figure S1A.
b/ This was misleading in our original version and we have now clarified this point in the Results section (see L220-227). In our pilot experiments to establish a triple cell type setting (NALM6/Raji cells, CAR T cells, monocytes), we found that IL-6 release from monocytes was already high when NALM6 cells and monocytes were co-cultured with untransduced (UT) T cells. We did not observe more IL-6 release when we co-cultured NALM6 cells and monocytes with CAR T cells. When we used Raji cells instead of NALM6 cells, the amount of IL-6 released was significantly different between UT T cells and CAR T cells. We have now included this data set in Figure S1F-G. We believe that this difference is likely due to a high alloreactivity of the T cells against NALM6 cells but not against Raji cells. Based on these experiments, we decided to use Raji cells in all subsequent experiments.
2 An analysis on CAR-T cells targeting other antigens is lacking to confirm the results.
Response:
It would indeed be interesting to extend our findings to other targets in the future. Based on the available clinical CAR T cell data, the incidence and severity of CRS appears to be higher in patients with hematologic malignancies than in patients with solid tumors. Furthermore, anti-CD19 CAR T-cell products induced a higher prevalence of CRS compared to anti-BCMA CAR T-cell products (80% vs. 4-5%, respectively). In addition, CD19-targeted CAR T cells are still one of the most commonly used CAR T cell products in the clinic. Therefore, we focused on CD19-targeting CAR T cells to establish our in vitro CRS models. The various control experiments we included, in particular CD19-negative target cells and UT T cells, helped us to validate the CRS in vitro model.
This clarification has been added to the Introduction (L43-47).
3 The in vitro potency of edited CAR-T cells is comparable to that of WT CAR-T. But I think data from in vivo experiments are also needed to validate these results.
Response:
It would certainly be interesting to study the activity of our engineered CAR T cell products in an in vivo model. However, the main focus of our work was to develop an in vitro model of CRS that contributes to the reduction of animal experimentation (3R principles). In this regard, our in vitro model allowed us to recapitulate some observations previously made in mouse models, such as the effect of GM-CSF knockout in CAR T cells on cytokine release without losing CAR T cell potency (see Sterner et al. who showed that the GM-CSF knockout CAR T cells had comparable potency to WT CAR T cells in an in vivo setting). To address the reviewer's concern, we have now more clearly stated in the Abstract (L31) and Discussion (L399-406, L428-430) that the functionality of our edited CAR T cells was assessed in vitro.
4 What do the authors think about CAR-T cells engineered to express CD40L and enhance immune response? (see Kuhn et al. Cancer Cell. 2019 Mar 18; 35(3): 473-488.e6). The CD40-CD40L axis seems to be implicated in enhancing the cytotoxic response.
Response:
We thank the reviewer for pointing out that we had not previously addressed this study. We have now cited the suggested paper and expanded our discussion (L407-423) to address the role of the CD40L-CD40R axis in the mouse and human immune system and why we need in vitro assays, such as the one we developed here, to better understand CRS and to further investigate genome editing strategies to mitigate CRS.
5 Finally, to my knowledge CRS is associated, according to a relevant part of the literature, with the presence of a large tumor and thus with the significant post-treatment presence of cells that have gone to death. What do the authors think about this?
Response:
We agree with the reviewer. There is a correlation between tumor burden and CRS severity. The pathophysiology of CRS is now better explained in the Introduction (L53-58; L79-82).
Reviewer 2 Report
Comments and Suggestions for Authors
In this article Dibas et al. documented that via an in-vitro system they can mimic the cytokine release syndrome (CRS) driven by CAR-T cell therapy. They focused on upregulation of CD40L and GM-CSF release as major targeting approach to mitigate the adverse effects of CAR-T cells. The study is very well written, and analyses were carefully performed. Although the design of the analysis was good, further clarifications are needed to justify the conclusions. The comments are as follows:
Comments:
1. Throughout the manuscript, the authors majorly focused on GM-CSF and IL-6 as their readout for CRS. The authors should show other cytokines in particular TNF-a, Granzyme B and IL-10 levels after the disruption of CD40L and GM-CSF? These will give a complete picture of the CRS.
2. Why the NALM6 did not activate the release of IL-6? Why the invitro system did not mimic the xenograft model of NALM6?
3. For Figure 1A and 1B the cartoons are not very clear and not very informative. The authors should improve the cartoons to clarify the experiments more clearly.
4. For Figure 2C and 2D, why the DKO has an opposite effect than the single KOs?
5. Since the authors did not getting a 100% CRISPR KO; if we culture the cells for longer time and perform the similar kind of experiments will that change the outcome? If in longer time course the non-deleted cells take over the deleted cells and yield a different results?
6. In the discussion/ conclusion section the authors did not clearly discuss how this strategy will advance our current knowledge and the future directions.
Comments on the Quality of English LanguageMinor changes required.
Author Response
Comments and Suggestions for Authors
In this article Dibas et al. documented that via an in-vitro system they can mimic the cytokine release syndrome (CRS) driven by CAR-T cell therapy. They focused on upregulation of CD40L and GM-CSF release as major targeting approach to mitigate the adverse effects of CAR-T cells. The study is very well written, and analyses were carefully performed. Although the design of the analysis was good, further clarifications are needed to justify the conclusions. The comments are as follows:
Response:
We thank the reviewer for his/her positive feedback. Below is our response to the reviewer, including the references to the new datasets provided.
Comments:
- Throughout the manuscript, the authors majorly focused on GM-CSF and IL-6 as their readout for CRS. The authors should show other cytokines in particular TNF-a, Granzyme B and IL-10 levels after the disruption of CD40L and GM-CSF? These will give a complete picture of the CRS.
Response:
We thank the reviewer for the suggestion to include these data. We did indeed look at the release of TNFa, IL-10, and INFg in our pilot triple culture experiments. We found that the released levels of these cytokines were similar in the co-culture experiments with edited vs. unedited CAR T cells (new Figure S3). The new data are referenced in the Results section (L361-363). In addition, we also looked at Granzyme B release, which is restricted to effector T cells rather than monocytes, in co-cultures of CAR T cells with CD19+ target cells. We observed no differences in granzyme B release between edited and unedited CAR T cells (Figure 4C).
- Why the NALM6 did not activate the release of IL-6? Why the invitro system did not mimic the xenograft model of NALM6?
Response:
We thank the reviewer for raising these points.
a/ This was misleading in our original version and we have now clarified this point in the Results section (see L220-227). In our pilot experiments to establish a triple cell type setting (NALM6/Raji cells, CAR T cells, monocytes), we found that IL-6 release from monocytes was already high when NALM6 cells and monocytes were co-cultured with untransduced (UT) T cells. We did not observe more IL-6 release when we co-cultured NALM6 cells and monocytes with CAR T cells. When we used Raji cells instead of NALM6 cells, the amount of IL-6 released was significantly different between UT T cells and CAR T cells. We have now included this data set in Figure S1F-G. We believe that this difference is likely due to a high alloreactivity of the T cells against NALM6 cells but not against Raji cells. Based on these experiments, we decided to use Raji cells in all subsequent experiments.
b/ Regarding NALM6 cells, we assume that there are some differences between our in vitro model and the mouse model. We have expanded the discussion to address this fact (L407-423).
- For Figure 1A and 1B the cartoons are not very clear and not very informative. The authors should improve the cartoons to clarify the experiments more clearly.
Response:
We agree that the cartoons are helpful for readers to follow the experimental setup. We have added more explanation to the legend of Figure 1 to make it easier for readers to follow the cartoons.
- For Figure 2C and 2D, why the DKO has an opposite effect than the single KOs?
Response:
Our data showed that the KO efficiency in the double KO experiments was lower compared to the single KO setup, both at the protein level (Figure 2C-D) and at the genomic level (Figure 2E-F). This observation is related to the fact that we transferred 4 CRISPR/Cas9 nucleases per sample in the double KO samples, as opposed to 2 nucleases for the single KO. To avoid cytotoxicity, we reduced the amount of each RNP per target in the DKO experiments to not exceed 6 µg of RNP per transfection. The setup is described in detail in the Materials and Methods section (L119-125). However, we have added a brief explanation in the Results section (L285-288).
- Since the authors did not getting a 100% CRISPR KO; if we culture the cells for longer time and perform the similar kind of experiments will that change the outcome? If in longer time course the non-deleted cells take over the deleted cells and yield a different results?
Response:
KO efficiency was typically determined 8 days after electroporation (Figure 2). In addition, in some experiments, edited CAR T cells were selected with CD271 antibodies and then expanded for at least 18 days before co-culture experiments were performed (Figure 4). Therefore, the evaluation of KO efficiency was performed at different time points, and it was stable throughout the time that the cells were kept in culture. We have clarified the timeline of the experimental setup in Materials and Methods (L129-131).
- In the discussion/ conclusion section the authors did not clearly discuss how this strategy will advance our current knowledge and the future directions.
Response:
We thank the reviewer for raising this point. We have now included a brief discussion of future directions (L449-457).
Round 2
Reviewer 1 Report
Comments and Suggestions for Authors
This revised version of the manuscript deserves to be published.
Reviewer 2 Report
Comments and Suggestions for Authors
The authors addressed my concerns. The manuscript can now be accepted for publication from my end.